# Induction of Bystander and Abscopal Effects after Electroporation-Based Treatments

**DOI:** 10.3390/cancers14153770

**Published:** 2022-08-02

**Authors:** Paulius Ruzgys, Diana Navickaitė, Rūta Palepšienė, Dovilė Uždavinytė, Neringa Barauskaitė, Vitalij Novickij, Irutė Girkontaitė, Brigita Šitkauskienė, Saulius Šatkauskas

**Affiliations:** 1Biophysical Research Group, Vytautas Magnus University, Vileikos St. 8, LT-44404 Kaunas, Lithuania; paulius.ruzgys@vdu.lt (P.R.); diana.navickaite@vdu.lt (D.N.); ruta.palepsiene@vdu.lt (R.P.); dovile.uzdavinyte@vdu.lt (D.U.); neringa.barauskaite@vdu.lt (N.B.); 2Faculty of Electronics, Vilnius Gediminas Technical University, Saulėtekio al. 11, LT-10223 Vilnius, Lithuania; vitalij.novickij@vilniustech.lt; 3Department of Immunology, State Research Institute Centre for Innovative Medicine, LT-08406 Vilnius, Lithuania; irute.girkontaite@imcentras.lt; 4Department of Immunology and Allergology, Medical Academy, Lithuanian University of Health Sciences, Eiveniu 2, LT-50161 Kaunas, Lithuania; brigita.sitkauskiene@kaunoklinikos.lt

**Keywords:** cell electroporation, bleomycin electrotransfer, calcium electroporation, electrochemotherapy, bystander effect, abscopal effect, IL-2 gene delivery, breast cancer

## Abstract

**Simple Summary:**

The delivery of electric field to tumor cells or nodules induces cell electroporation that allows intracellular delivery of cytotoxic agents and eventually inhibits tumor growth. In this study, we showed that intracellular delivery of calcium ions and anticancer drug bleomycin not only kills the cells but also has a negative bystander effect on indirectly treated cells. We also showed that, when directly applied to one tumor, these treatments can inhibit the growth of a second, non-electroporated tumor.

**Abstract:**

Electroporation-based antitumor therapies, including bleomycin electrotransfer, calcium electroporation, and irreversible electroporation, are very effective on directly treated tumors, but have no or low effect on distal nodules. In this study, we aimed to investigate the abscopal effect following calcium electroporation and bleomycin electrotransfer and to find out the effect of the increase of IL-2 serum concentration by muscle transfection. The bystander effect was analyzed in in vitro studies on 4T1tumor cells, while abscopal effect was investigated in an in vivo setting using Balb/c mice bearing 4T1 tumors. ELISA was used to monitor IL-2 serum concentration. We showed that, similarly to cell treatment with bleomycin electrotransfer, the bystander effect occurs also following calcium electroporation and that these effects can be combined. Combination of these treatments also resulted in the enhancement of the abscopal effect in vivo. Since these treatments resulted in an increase of IL-2 serum concentration only in mice bearing one but not two tumors, we increased IL-2 serum concentration by muscle transfection. Although this did not enhance the abscopal effect of combined tumor treatment using calcium electroporation and bleomycin electrotransfer, boosting of IL-2 serum concentration had a significant inhibitory effect on directly treated tumors.

## 1. Introduction

Local delivery of electric field pulses to the tumor cells or tumor nodules may lead to reversible electroporation (RE) or irreversible electroporation (IRE) of tumor cells [1,2,3]. Both types of electroporation have been systemically investigated and applied for in vivo preclinical and clinical studies as novel modalities of antitumor treatment [2,3,4,5,6,7]. To achieve antitumor response with RE, anticancer drugs (e.g., bleomycin or cisplatin) or calcium are administered prior to delivery of electric pulses, either locally or systemically. These antitumor therapies are known as antitumor electrochemotherapy and calcium electroporation, respectively [6,8,9]. Currently, ECT is a routine clinical treatment, reported to be a highly effective and safe treatment when European Standard Operating Procedures of Electrochemotherapy (ESOPE) are followed [10]. To this date, bleomycin is the most frequently used anticancer drug for ECT, and intravenous (i.v.) injection is the most common administration route for this drug [11]. In some cases, cisplatin can be very effective in ECT and can be used as an alternative of bleomycin [12]. Recently, a new modality of RE combined with intratumoral administration of supraphysiological doses of calcium ions, calcium electroporation, has been implemented in clinical trials [8,9].

With IRE, an antitumor effect can be achieved without use of any cytotoxic compound or anticancer drug. IRE alone causes dramatic changes in membrane and subsequently cellular homeostasis leading to cell death [1,3,13]. Many studies have demonstrated that IRE can be applied in vivo as nonthermal tissue ablation and tumor treatment [7,14]. Such electroporation-based tumor modality has been extensively explored in clinical trials, showing increasing promises for IRE to be applied for treatment against certain types of cancer in clinic [15,16,17,18,19].

Although electroporation-based antitumor therapies are highly efficient locally, a lack of lack or a low systemic response hinders broader application of the treatment. Some attempts have been made to activate systemic response by combining ECT with immunomodulatory molecules, such as interleukins IL-2, IL-12, tumor necrosis factor alpha, interferon alpha, granulocyte–macrophage colony-stimulating factor [20,21,22,23,24], or antibodies targeting the immune system’s cells (ipilimumab or pembrolizumab) [25]. It has also been reported that calcium electroporation results not only in efficient tumor eradication following local treatment, but also induces long-lasting immunity and cytokine responses [26]. Similarly, it has been shown that IRE-treated tumors change the status of cellular immunity and may invoke systemic response beyond the targeted ablation region [27,28,29]. A growing number of studies related to IRE-based tumor treatments reveal that the combination of IRE and immunotherapy may lead to long-term memory immune response and overall enhancement of necrosis in the off-target tumors [30,31,32]. Many recent clinical studies related to IRE have been initiated to delineate further perspectives of IRE-based tumor treatments and opportunities for synergistic therapies [15,33,34].

Synthesis and release of damage-associated molecular patterns (DAMPs) [30,35,36] following electroporation is the most common explanation of how electroporation-based cancer treatments can induce immune responses. In our recent study, we demonstrated that bleomycin electrotransfer can induce a negative bystander effect on cells that were not treated directly [37]. Interestingly, we showed that irreversible electroporation can trigger a positive bystander effect in vitro, presumably because of cell viability promoting factors released in the medium from irreversibly electroporated cells. It remains unclear how these bystander effects could contribute to antitumor response following electroporation-based therapies.

In this study, we performed additional research to analyze the bystander effect after bleomycin electrotransfer and calcium electroporation in vitro. We further explored whether these treatments can induce an abscopal effect in vivo alone or in combination. To enhance possible systemic response and antitumor effect of the indirectly treated tumors, we aimed to boost IL-2 serum level by electrotransfer of IL-2-coding plasmid into tibialis anterior muscles of the treated mice.

## 2. Materials and Methods

### 2.1. In Vitro Experiments and Cell Preparation for Tumor Injections

All in vitro experiments were performed using 4T1 cells. The 4T1 cell (CRL-2539) line was acquired from ATCC (American Type Culture Collection, Manassas, VA, USA). https://www.atcc.org/products/crl-2539 (accessed 20 October 2021). This tumor-derived cell line is an animal model for stage IV human breast cancer. The cells were grown in RPMI media supplemented with 10% of fetal bovine serum and 1% of penicillin/streptomycin solution. Cells were re-cultured every third day and had less than 80% of confluency on the surface of the Petri dish or 24 h before every experiment.

For the subcutaneous injection in mice, cells were detached from the Petri dish with TrypLE (Gibco) centrifuged and resuspended in saline solution (0.9% NaCl) at concentration of 20 × 10^6^ cells/mL. The injections in mice were performed during 30 min after the preparation of 4T1 cell suspension.

#### 2.1.1. In Vitro Cell Electroporation

After trypsinization using trypsin-EDTA solution (Sigma-Aldrich, St. Louis, MO, USA), the cells were suspended in a laboratory-made electroporation buffer (Na_2_HPO_4_ 5.59 mM, NaH_2_PO_4_ 3.00 mM, MgCl_2_ 1.73 mM, sucrose 242.2 mM) at a concentration of 2 × 10^6^ cells/mL. The conductivity of the electroporation media was 0.1 S/m with pH 7.3. Afterward, 45 μL of prepared cell suspension was mixed with 5 μL anticancer drug bleomycin (200 nM) or with 5 μL of CaCl_2_ (10 mM) suspension. The resulting 50 μL of cell suspension containing 9 × 10^5^ of cells and 20 nM of bleomycin or 1 mM of CaCl_2_ was then transferred in between laboratory-made stainless steel plate electrodes with a 2 mm gap and electroporated using one 100 µs duration square wave pulse ranging from 600 to 1400 V/cm strength using a BTX T820 electroporator (Harvard Apparatus, San Diego, CA, USA).

#### 2.1.2. In Vitro Cell Viability Evaluation of Cells Directly Treated with Bleomycin Electrotransfer or Calcium Electroporation

After the application of electric fields, the treated cell suspension was placed in a 1.5 mL Eppendorf tube and incubated for 10 min. Then, 400 of cells from the suspension were plated in a 40 mm Petri dish (TPP) with 2 mL growth medium and transferred into the incubator (5% CO_2_ at 37 °C) for 6 days. Afterward, formed colonies were stained with crystal violet solution (40% ethanol, 20% distilled water, 40% crystal violet dye (Sigma-Aldrich, St. Louis, MO, USA)). The number of the colonies was evaluated using ImageJ software (National Institute of Health, Bethesda, MD, USA) following recommendations provided by the creators of the software [38,39] and compared to the number of the colonies in the control.

#### 2.1.3. In Vitro Cell Viability Evaluation of Cells Indirectly Treated with Cell Medium Taken from the Directly Treated Cells (Evaluation of Bystander Effect)

After bleomycin electrotransfer or calcium electroporation, the cell suspension was placed in a well of a 24-well plate (TPP) and supplemented with 0.2 mL of RPMI growth medium after 10 min of incubation. The same experiment was repeated to fill 10 wells of the 24-well plate. Afterward, the 24-well plate was transferred into the incubator (5% CO_2_ at 37 °C) for 48 h. Then, the conditioned medium (CM) was collected, centrifuged twice for further use. Therefore, CM composed of 20% of electroporation media and 80% of growth medium taken from treated cells. A total of 2 mL of CM (undiluted or diluted with a freshly prepared growth medium) was transferred to previously (24 h before) plated, untreated 400 cells in a 40 mm Petri dish (TPP). The Petri dishes were then transferred into an incubator (5% CO_2_ at 37 °C) for 5 days. Afterward, the colonies formed were evaluated as described above.

After 10 min post electroporation, the intracellular propidium iodide (PI) was measured with flow cytometer (BD Accuri C6, Franklin Lakes, NJ, USA). For all measurements, 22 mm core size and 66 mL/min flow rate were used. In total, 10^4^ cells per sample were measured. The flow cytometry gating strategy to collect PI-positive cells is presented in Figure 1. In short, three segments were marked, with the leftmost representing viable cells, the middle one representing PI-positive viable cells, and the rightmost representing PI-positive dead cells. As shown in Figure 1A, only a small number (about 2.9%) of PI-stained cells can be detected in the control (any treatment) sample. Meanwhile, electroporation using 800 or 1400 V/cm strength pulses resulted in a significant shift of fluorescence and a varying number of PI-positive cells (51.8 and 74.1%, respectively) (Figure 1B,C). PI-positive cell values were taken from FL-2 channel (excitation 488 nm and emission 585/40 nm) fluorescence values. Data analysis was performed using FlowJo Single-Cell Analysis Software, Becton Dickinson, (Ashland, OR, USA).

### 2.2. In Vivo Experiments

The in vivo experiments were approved by the Lithuanian State Food and Veterinary Service (2020-05-05 number Nr. G2-149). Eight to twelve-week-old female BALB/c mice were used in the experiments. The mice were purchased from Department of Biological Models, Institute of Biochemistry, Vilnius University (Vilnius, Lithuania). Six mice per cage were housed in plastic cages with ad libitum access to water and food. They were housed under constant environmental conditions with a 12-h light-dark cycle. The induction of the tumor was performed by subcutaneous injection of 4T1 cells on one or two dorsal caudal sides of the mice. One 100 µL of volume injection contained 4T1 cells at a concentration of 20 × 10^6^ cells/mL diluted in saline solution. Either one or two tumors were induced on a single mouse, based on assignment to specific experimental group. Depending on the tumor growth rate, the experiments were performed at the period between day 9 and day 12. One day before the experiment, all mice were shaved, and the remaining hair was removed with depilation cream.

#### 2.2.1. Electrochemotherapy and Calcium Electroporation

Calcium electroporation, bleomycin electrotransfer, or the combination of both was performed on sedated mice. For anesthesia, a ketamine/xylazine solution was used at concentrations 87.5 and 12.5 mg/kg, respectively. For antitumor treatment using bleomycin, 200 µL (diluted in saline solution) of the drug at 470 µM concentration was injected into the mouse tail vein. Electric fields were applied 10 min after the injection. For calcium electroporation, an amount of half tumor size volume of CaCl_2_ solution (168 mM) was injected intratumorally (i.t.), following the protocol described in [40]. The measurement of tumor volume is described in the ‘Tumor size measurements’ section. Electric fields were applied 10 min after the injection. Both procedures were performed in case of simultaneous bleomycin and CaCl_2_ electrotransfer.

#### 2.2.2. Electrotransfection

For the electrotransfection experiments, pUNO1-mIL02 (InvivoGen, Toulouse, France) plasmid coding for mouse interleukin 2 was used. The plasmid was propagated in *E. coli* DH5α and purified using the EndoFree Plasmid GigaPrep kit (Qiagen, Chatsworth, CA, USA) according to the manufacturer’s instructions. The extracted plasmid DNA was validated by electrophoresis on agarose gel. The concentration and purity were determined by using Nanodrop 2000 (Thermo Fisher, Washington, DC, USA).

Electrotransfection was done on both tibialis anterior muscles of each treated mouse. Firstly, the back legs of the mice were shaved, and remaining hair was removed with depilation cream. Afterward, 35 µL of the IL-2 coding plasmid solution (diluted in saline solution) at a concentration of 1 mg/mL was injected into the tibialis anterior muscles of sedated mice. Electric fields on the muscle were applied 10 min after the injection. One or both tibialis anterior muscles were electro-transferred during the treatment.

#### 2.2.3. Electroporation Parameters

For in vitro experiments, single 100 µs duration pulse at pulse strength ranging from 600 to 1500 V/cm was used. For in vivo experiments, stainless steel electrodes mounted on the caliper were used. A conductive gel was applied to the electrodes to ensure good contact between electrode plates and skin. The electrodes were mounted on the tumor, and the distance between electrodes was evaluated before adjusting and delivering electric field pulses. Tumors were treated with eight 100 μs duration square-wave electric pulses at 1200 or 1500 V/cm strength and 1 Hz frequency, following ESOPE or ESOPE-like protocol [10]. For IL-2 coding plasmid electrotransfer, tibialis anterior muscles were electroporated using eight 20 ms duration square-wave electric pulses at 200 V/cm strength and 1 Hz frequency, which were applied 10 min prior to the injection of the plasmid.

#### 2.2.4. Tumor Size Measurements

The tumors were measured daily, starting on day 1 (day of the experiment) and finishing on day 11. The evaluation of tumor volume (V) was done by measuring the length, width, and height of the tumor and calculated according to equation V = π × l × w × h/6. All the measurements were performed using an electronic caliper.

#### 2.2.5. Serum Preparation and ELISA

On the last day of the experiments, the mice were euthanized according to the guidelines for euthanasia of rodents using carbon dioxide. The chest cavity of each euthanized mouse was opened, and the maximum available volume of blood was collected from the heart. The serum was collected by centrifugation of the blood at 1000× *g* for 3 min and stored at −20 °C for ELISA experiments. The concentration of IL-2 in serum was done using DuoSet ELISA system, and the measurements were performed according to the guidelines offered by DuoSet ELISA development system. The absorbance of the color at 450 nm was measured with plate reader TECAN Genios pro (Männedorf, Switzerland).

### 2.3. Statistical Analysis

Statistical analysis was performed with MS Excel and Prism 9 software. The data in the figures are represented as mean ± standard error of the mean (SEM). Statistically significant differences between experimental groups were evaluated using a *t*-test. The significance was marked as *, **, or *** when the *p*-values were less than 0.05, 0.01, or 0.001, respectively. There were at least 3 independent experiments with 3 repeats of each experimental point in in vitro experiments, and at least 8 individual mice in in vivo experiments used for each experiment point.

## 3. Results

The first experiment was designed to determine the optimal electric field strength for reversible electroporation of 4T1 line cells in vitro using single 100 µs duration electric pulse. We measured both percentage of PI-positive cells as well as cell viability (Figure 2A). We chose 1400 V/cm pulse strength for further studies as at this strength percentage of PI-positive cells was close to the highest, 87%, while decrease in cell viability to 84.4% was still not significant.

Further, we investigated cell viability after bleomycin (20 nM) electrotransfer and calcium (1 mM) electroporation. These treatments at 1400 V/cm resulted in decrease of cell viability to 21.36 and 13.35%, respectively (Figure 2B).

In a recent study, we demonstrated the presence of the bystander effect after bleomycin electrotransfer in CHO cells [37]. In the present study, we aimed to build on the previous results and analyze the bystander effect on 4T1 cells after bleomycin electrotransfer and after calcium electroporation. To this aim, we treated cells with a CM taken at various time points from the untreated cells, cells exposed to bleomycin electrotransfer, or to calcium electroporation (Figure 3A). Notably, medium taken from untreated cells (control) had a minor effect on cell viability, depending on the time point of medium collection. When the CM was collected at 48 h after the experiment, the viability of cells grown in CM remained close to 70%. Cell treatment with CM collected at this time point from cells exposed to bleomycin electrotransfer (BLM + EP) or to calcium electroporation (CaCl_2_ + EP) resulted in the death of all indirectly treated cells (Figure 3A). Since CM collected at 48 h had significant bystander effect (indirect treatment) on cell viability, further experiments were performed with CM collected at this time point.

We further analyzed the bystander effect after CM dilution with fresh growth medium (50% of CM medium and 50% fresh growth medium). As expected, dilution of CM from cells exposed to bleomycin electrotransfer or to calcium electroporation resulted in an increase of cell viability from 0% to 69.71% (Figure 3B) and 94.9% (Figure 3C), respectively. Surprisingly, when CM collected from cells exposed to both bleomycin electrotransfer and calcium electroporation were diluted to 50% and mixed, it still resulted in death of all cells (Figure 3D).

After observation of a significant influence of CM medium collected from cells exposed to bleomycin electrotransfer or calcium electroporation on cell viability, when used alone or in combination, we decided to evaluate whether similar effect could be observed in in vivo experiments. For the initial experiments, mice with a single tumor were subjected to treatment using eight 1200 V/cm, 100 µs duration electric field pulses according to ESOPE protocol [10] prior to administration of bleomycin, CaCl_2_, or both (Figure 4).

There were no significant differences in cancer growth rate when CaCl_2_ or bleomycin were administered separately without electroporation, or when only electric pulses were delivered to the tumor. As expected, both bleomycin electrotransfer and calcium electroporation resulted in a very similar and statistically significant inhibition of tumor growth; tumor volume in these treated groups remained stable (40–110 mm^3^) during the period of 10 days and significantly lower compared to the untreated control (502 mm^3^) (*p* < 0.001). However, combination of bleomycin electrotransfer and calcium electroporation resulted in insignificant tumor growth decrease when compared to bleomycin electrotransfer or calcium electroporation alone (*p* > 0.05).

Furthermore, we aimed to evaluate the response to the treatment of primary tumor using bleomycin electrotransfer, calcium electroporation, or a combination of both on indirectly treated tumors. This indirect effect has been commonly observed after application of ionizing radiation and is called abscopal effect [41,42]. To investigate the abscopal effect, we inoculated two tumors on both flanks of the mice. One tumor was treated by EP-based therapies (directly treated tumor), and another was left untreated (indirectly treated tumor) (Figure 5A). Initially we tested the effect of administration of bleomycin, CaCl_2_, or use of electroporation on directly treated tumors and indirectly treated tumors (Figure 5B,C). Electroporation alone or i.t. injection of CaCl_2_ alone resulted in slight but significant reduction of tumor growth compared to the control (*p* < 0.05). These results were unexpected as CaCl_2_ injection alone did not have any effect when the experiments were performed on mice bearing one tumor. On the other hand, no significant tumor growth decrease was observed in the untreated tumor group. BLM i.v. injection did not result in any significant inhibition of tumor growth.

We further investigated the abscopal effect, i.e., antitumor effect on indirectly treated tumor after bleomycin electrotransfer, calcium electroporation, or combination of both (Figure 6). Tumors were electroporated using 1200 V/cm (A and B panels) and 1500 V/cm electric field pulses (C and D panels). Bleomycin electrotransfer, calcium electroporation, or combination of both using 1200 V/cm pulses significantly inhibited growth of directly treated tumors. At the end of the 11-day evaluation period, tumor sizes (149, 189, and 90 mm^3^, respectively) were significantly smaller compared to the control (502 mm^3^; *p* < 0.001). Moreover, results revealed that size of the indirectly treated tumors significantly decreased by ~50%: from 502 mm^3^ in the control to 252 and 214 mm^3^ after bleomycin electrotransfer or combination of bleomycin electrotransfer and calcium (*p* < 0.001). Interestingly, calcium electroporation alone had no significant effect on growth of indirectly treated tumors (Figure 6B).

To find out whether the growth of directly and indirectly treated tumors can be inhibited further, we used stronger, 1500 V/cm electric field pulses. When tumors were treated with these pulses (in presence of bleomycin, CaCl_2_, or both), almost complete reduction of tumors was observed in all groups of directly treated tumors (Figure 6C). Tumor sizes at the end of the 11-day evaluation period were 35, 32, and 11 mm^3^ after bleomycin electrotransfer, calcium electroporation, or combination of both. Significant (*p* < 0.001) inhibition of tumor growth was also observed in the indirectly treated tumors. In this case, bleomycin electrotransfer or combination of bleomycin electrotransfer and calcium electroporation inhibited tumor growth, resulting in mean tumor volume at the end of the 11-day evaluation period of 218 and 155 mm^3^, respectively. Importantly, calcium electroporation alone also resulted in significant inhibition of tumor growth (*p* < 0.001).

Even though the biological mechanism behind the abscopal effect is still not fully understood, the immune system is believed to play a key role. Therefore, the next set of experiments was designed to evaluate the immune system response to the electroporation-based treatments. We have chosen IL-2 as a marker for immune system response to the treatment. Therefore, we measured changes of IL-2 concentrations in mice serum collected on the last day of the experiment (Figure 7).

Evaluation of IL-2 concentration after electroporation at 1200 V/cm alone or CaCl_2_ injection alone revealed no significant difference from the control (Figure 7A). However, both bleomycin electrotransfer and calcium electroporation alone, or a combination of both, resulted in a significant increase in IL-2 concentration as shown in Figure 7B. IL-2 concentration increased from 21.5 pg/mL in the control group to 34.8, 32.8, and 58.5 pg/mL after bleomycin electrotransfer, calcium electroporation, or combination of both, respectively (*p* < 0.01). Surprisingly, the effect was observed only in mice with a single tumor.

Since previously described results revealed that 1500 V/cm was more effective in inhibiting tumor growth, we expected to detect higher IL-2 concentration as a response. Indeed, results presented in Figure 7C revealed statistically significant IL-2 concentration increase after bleomycin electrotransfer (*p* < 0.01) or combination of calcium electroporation and bleomycin electrotransfer (*p* < 0.05) when an electric pulse of 1500 V/cm was used. This suggests a direct connection between increase of IL-2 concentration in serum and the strength of tumor growth inhibition.

To test this hypothesis, we decided to increase the IL-2 concentration by transfecting both tibialis anterior mouse muscles with IL-2 gene coding plasmid (pUNO1-mIL02). IL-2 concentration in the bloodstream was then evaluated before the treatment and at days 1, 3, 7, 11, and 14 (Figure 8A). The results showed that following IL-2 coding plasmid electrotransfer, IL-2 concentration significantly increased from 19 pg/mL in the control (untreated muscles) to 258 and 307.5 pg/mL on the first and third days after plasmid electrotransfer, respectively (*p* < 0.01). Afterward, IL-2 concentration gradually decreased and was similar to the control at day 14. Notably, muscle transfection by IL-2 coding plasmid also inhibited tumor growth (Figure 8B). Indeed, a significant (*p* < 0.001) inhibition of tumor growth was observed: tumor size at the end of the 11-day evaluation period reached 502 mm^3^ in the control group and 186 mm^3^ in the IL-2 group.

Since a boost of serum IL-2 concentration following IL-2 plasmid electrotransfer resulted in significant growth inhibition of indirectly treated tumors (abscopal effect) (Figure 8B), we further aimed to analyze the effect of IL-2 on tumor growth following electroporation-based treatments (Figure 9). IL-2 transfection following calcium and bleomycin electrotransfer resulted in significant inhibitory effect on directly treated tumors (Figure 9A); however, it had no additional effect on indirectly treated tumors (Figure 9B). Therefore, it seems that overexpression of IL-2 does not enhance the abscopal effect of combined tumor treatment using calcium electroporation and bleomycin electrotransfer; however, it has a significant inhibitory effect on the growth of directly treated tumors.

To further understand the effect of different treatment strategies on the tumor growth dynamics, we decided to evaluate the number of tumors with complete response. Complete responses were observed only in a directly treated tumors. When combined treatment strategies were applied (Figure 9C), a combination of IL-2 transfection and both calcium electroporation and bleomycin electrotransfer using 1500 V/cm electrical pulses proved to be the most effective and resulted in complete responses in 100% of directly treated tumors. A combination of calcium electroporation and bleomycin electrotransfer without IL-2 transfection resulted in complete response in 40% of directly treated tumors (Figure 9C).

In the last set of experiments, we aimed evaluate how different electroporation-based treatments alone or in combination with IL-2 transfection affect IL-2 concentrations in the serum collected at the last day of the 11-day evaluation period (Figure 10). As expected, IL-2 transfection resulted in increase of IL-2 concentration compared to the control, from 20.3 pg/mL in the control to 48.7 pg/mL in IL-2 transfection group (*p* < 0.01). Some increase in IL-2 concentrations was also observed in bleomycin electrotransfer and calcium electroporation groups; however, a significant difference from the control was observed only when these treatments were combined (*p* < 0.05).

## 4. Discussion

In our recent study, we demonstrated the negative bystander effect on CHO cells after bleomycin electrotransfer and hypothesized that the negative bystander effect could be triggered by a similar mechanism to the bystander effect induced by ionizing irradiation [37,43,44]. In this study, we extended analysis of the bystander effect and for the first time showed the presence of bystander effect on 4T1 cells after calcium electroporation (Figure 2). The bystander effect disappeared when CM was diluted with growth medium. However, when diluted CM from calcium electroporated cells was mixed with diluted CM from cells treated with bleomycin, the negative bystander effect was restored (Figure 3).

Combination of the bystander effect in vitro induced by bleomycin electrotransfer and calcium electroporation suggested a possibility of a combination of these effects in vivo. We also hypothesized that in in vivo settings, the bystander effect can operate not only on neighboring cells but also modulate the immune response and induce an abscopal effect as suggested by other studies [45,46]. Indeed, we have observed the abscopal effect of combined bleomycin electrotransfer and calcium electroporation treatment both at 1200 and 1500 V/cm pulse application (Figure 6). These results also showed that the abscopal effect induced by bleomycin electrotransfer was stronger than that induced by calcium electroporation, since the latter was absent when 1200 V/cm pulses were used.

The mechanism of action of the abscopal effect is closely related to the immune system of the treated organism [45]. Since IL-2 is known to increase sensitivity and proliferation of natural killer (NK) cells and cytotoxic T cells [47], we decided to monitor changes of this cytokine concentration in serum following abscopal effect inducing electroporation-based treatments. Additional reason for selecting IL-2 cytokine was the existing literature on its effect in electroporation-based treatments [48,49,50].

Our results showed an increase in serum IL-2 concentration following bleomycin electrotransfer, calcium electroporation, or combination of both; surprisingly, this was only observed in mice bearing one but not two tumors (Figure 7). The nature of this effect is not known and is worth to be analyzed in future studies. Encouraged by these results, we decided to move forward and increase IL-2 concentration in the serum via IL-2 gene coding plasmid transfection. Monitoring of IL-2 serum concentration following muscle transfection revealed that the of IL-2 concentration peaked on third day following transfection and decreased to the background level 10 days later (Figure 8). Presumably, this decrease can be explained by short half-life of IL-2 [51]. Interestingly, IL-2 expression in the muscle resulted in abscopal effect comparable to bleomycin electrotransfer and calcium electroporation (Figure 8B). This seems to suggest that the abscopal effect after bleomycin electrotransfer and calcium electroporation is associated with increased IL-2 concentration in the serum and may explain the results presented in Figure 7.

Since we observed a boost of serum IL-2 concentration following muscle transfection, we expected to find an increase in the magnitude of abscopal effect combining IL-2 transfection with bleomycin electrotransfer and/or calcium electroporation. Although we observed an expected effect on directly treated tumors (both on inhibition of tumor growth and increase in complete responses to treatment), we did not find any changes of the abscopal effect on indirectly treated tumors. This may suggest that IL-2 signaling pathway meets a bottleneck restricting the augmentation of the abscopal effect. It is known that IL-2 enhances the proliferation and differentiation of T cells from naïve to effector T-cells [52,53,54] and is a growth factor of NK cells [55] while additionally stimulating NK cell activation and toxicity [56,57]. On the other hand, elevated levels of IL-2 may also induce a selective expansion of regulatory T-cells that limits the activity of NK cells, resulting in poor clinical responses to IL-2 therapy [58]. Additionally, a high dose of IL-2 makes T-cells (particularly CD4+, which produce IL-2) short-lived [48]. This indicates that high levels of IL-2 can decrease the production of IL-2 from the immune cells. As the abscopal effect is directly dependent on the immune response, high amounts of IL-2 can suppress the natural elevation of IL-2 expressions as presented in Figure 7. To overcome the possible self-inhibitory effect of high IL-2 concentrations, we performed preliminary experiments by transfecting tibialis anterior muscles sequentially (one leg first and the second later) with 4 days’ time-lag between transfections. Nevertheless, it seems that this setting had no additional effect in enhancing the observed abscopal effect (data not shown).

While the abscopal effect was not enhanced by IL-2 overexpression, it had significant antitumor effect on directly treated tumors (Figure 9). This can be related with the fact that electrochemotherapy exerts immune response by inducing both immunogenic cell death and proliferation of regulatory T-cells [59]. Moreover, it has been shown that ATP is found in the extracellular matrix after electroporation [35,60]. ATP has been shown to recruit a range of immune effector cells, including dendritic cells (DC) and their precursors, to maintain these cells in the proximity of treated tumor cells and to promote DC maturation [61]. Indeed, it has been reported an extensive infiltration of T cells and macrophages into the treated tumors, as compared with the untreated tumors, at 48–72 h after low electric field-enhanced chemotherapy of melanoma in mice [62]. Similarly, electrochemotherapy with cisplatin was found to induce significant intratumoral recruitment of B cells, NK cells, DCs, macrophages and neutrophils [63]. Some of these effects could possibly be enhanced by IL-2 overexpression. In addition, IL-2 overexpression could enhance antitumor response of directly treated tumors via release of tumor-associated antigen and consequent activation of T-cells [64]. Nevertheless, a clear mechanism of these effects is unknown, showing the need of more systemic investigations. Possible directions can be drawn from very recent reviews on interaction of EP-based therapies and immune response [45,65].

## 5. Conclusions

In this study, we showed the presence of the bystander effect following calcium electroporation in vitro. Since a similar bystander effect is present following bleomycin electrotransfer, we showed that these treatments can be combined both in vitro and in vivo to enhance bystander and abscopal effect. Boosting of IL-2 serum concentration did not enhance the abscopal effect of combined tumor treatment using calcium electroporation and bleomycin electrotransfer; however, it had a significant inhibitory effect on directly treated tumors.

## Figures and Tables

**Figure 1 cancers-14-03770-f001:**
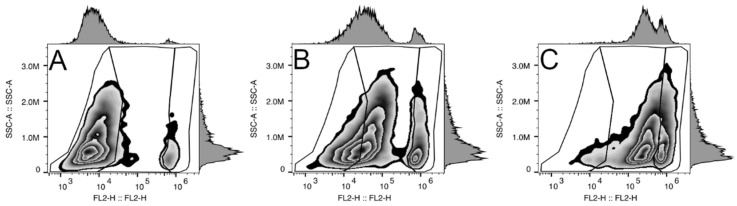
Gating strategy for the determination of successful PI electrotransfer and dead cells. (**A**) represents control cells (not treated with electric fields); (**B**,**C**) represents cells electroporated with one 100 µs-duration pulse at 800 V/cm and 1400 V/cm electric field strength, respectively. The Y axes show side scatter, and the X axes show the values of FL-2 channel (excitation 488 nm, emission 585/40 nm) fluorescence values. Three segments are marked: leftmost: viable cells, middle: PI-positive viable cells, rightmost: PI-positive dead cells.

**Figure 2 cancers-14-03770-f002:**
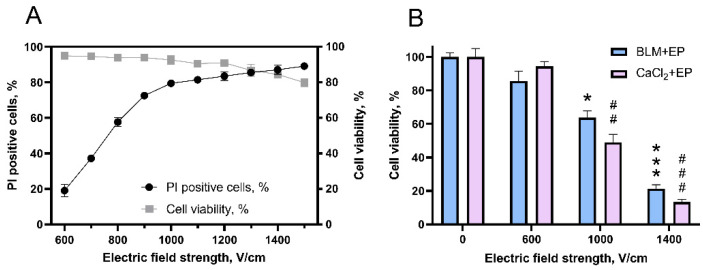
Viability decrease in 4T1 cells induced by bleomycin electrotransfer and calcium electroporation in dependence of electric pulse strength after cell electroporation using a single 100 µs duration pulse. (**A**) The dependence PI electrotransfer and cell viability on electric field pulse strength. (**B**) The dependence of clonogenic cell viability on electric field pulse strength after bleomycin (20 nM) electrotransfer or calcium (1 mM) electroporation. The mean represents 3 independent experiments with 3 repetitions. * and *** symbols represent statistical significance of comparison between BLM + EP group and control (cell not treated with electric pulses), respectively (*p* < 0.05 and *p* < 0.001). ## and ### symbols represent statistical significance of comparison between CaCl_2_ + EP group and control, respectively (*p* < 0.01 and *p* < 0.001).

**Figure 3 cancers-14-03770-f003:**
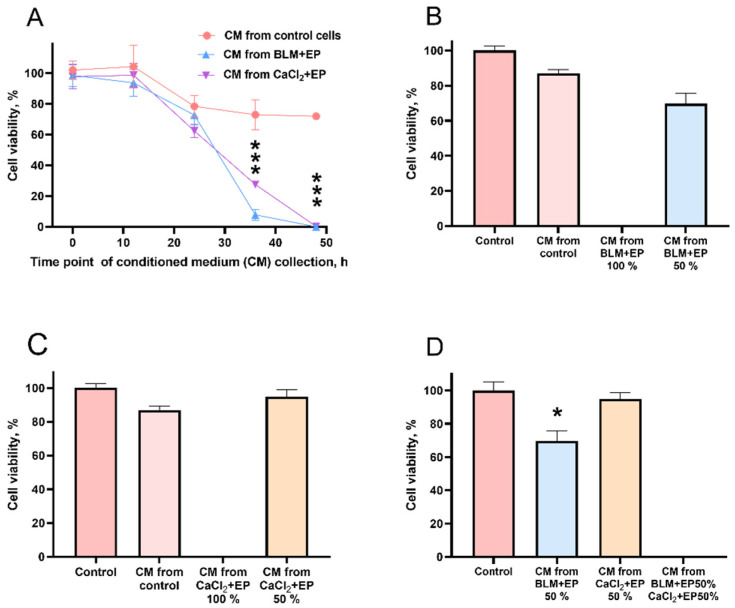
The bystander effect on 4T1 cells induced by bleomycin electrotransfer and calcium electroporation. (**A**) Cell viability after cell incubation in CM taken from untreated cells and cells exposed to (**B**) bleomycin (20 nM) electrotransfer or (**C**) calcium (1 mM) electroporation using a single 100 µs duration pulse at 1400 V/cm pulse strength. (**D**) Cell viability depending on CM dilution and mixing. The mean represents 3 independent experiments with 3 repetitions. * and *** symbols represent statistical significance of comparison between corresponding experimental group and control, respectively (*p* < 0.05 and *p* < 0.001).

**Figure 4 cancers-14-03770-f004:**
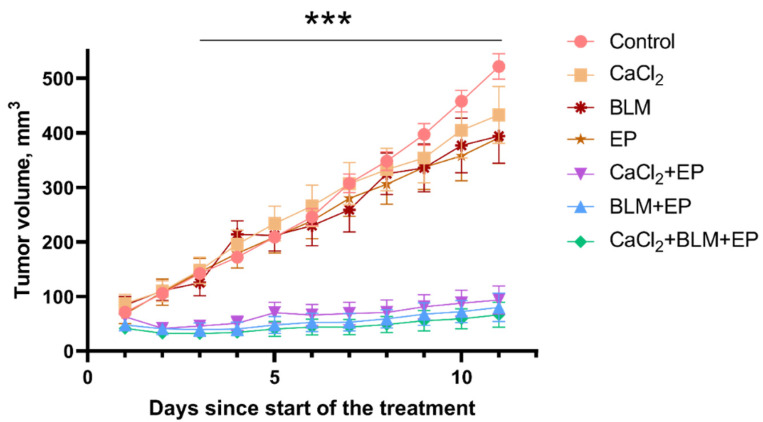
Tumor growth changes in controls and after treatment using bleomycin electrotransfer, calcium electroporation, or the combination of both. Bleomycin solution (200 µL, 470 µM) was injected in the mouse tail vein and CaCl_2_ solution (half of the tumor volume, 168 mM) was injected directly into the tumor 10 min prior application of electric fields. Eight 1200 V/cm, 100 µs duration pulses were used for the tumor electroporation. At least 8 mice bearing one tumor each were used for each experiment point. Statistical significance of differences between controls and experimental groups were evaluated using *t*-test. *** symbols represent statistical significance of *p* < 0.001 for comparison between control (no treatment) and individual experimental groups (CaCl_2_ + EP, BLM + EP, and CaCl_2_ + BLM + EP).

**Figure 5 cancers-14-03770-f005:**
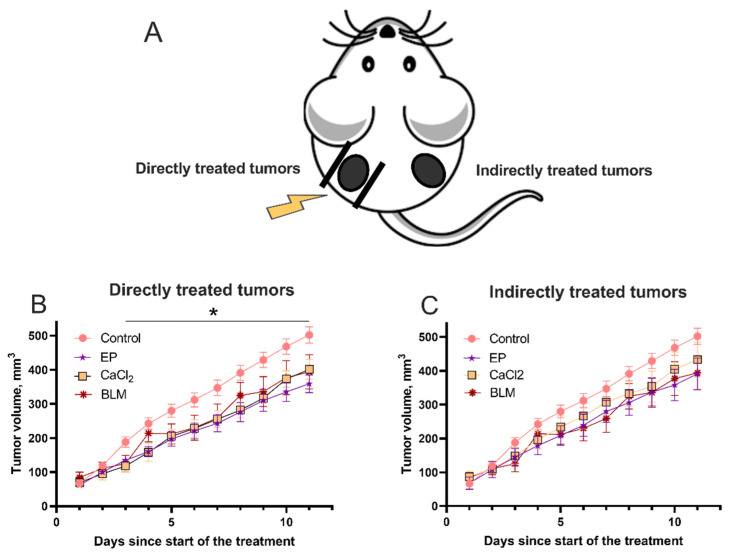
(**A**) An experimental setup of two tumors in one mouse to observe an abscopal effect. The electroporated or injected tumor is termed ‘directly treated tumor’, and the tumor that was directly untreated is termed ‘indirectly treated tumor’. Tumor growth changes in (**B**) treated tumor using electroporation or i.t. injection of CaCl_2_ solution and (**C**) untreated tumor. Eight 1500 V/cm, 100 µs duration pulses were used for the tumor electroporation. CaCl_2_ solution (half of the tumor volume, 168 mM) was injected directly into the tumor. At least 8 mice with one tumor in each were individually treated for every experiment point. Statistically significant differences between individual experimental groups were evaluated using *t*-test. * symbol represents statistical significance (*p* < 0.05) of comparison between EP and CaCl_2_ treatments versus control (no treatment), respectively.

**Figure 6 cancers-14-03770-f006:**
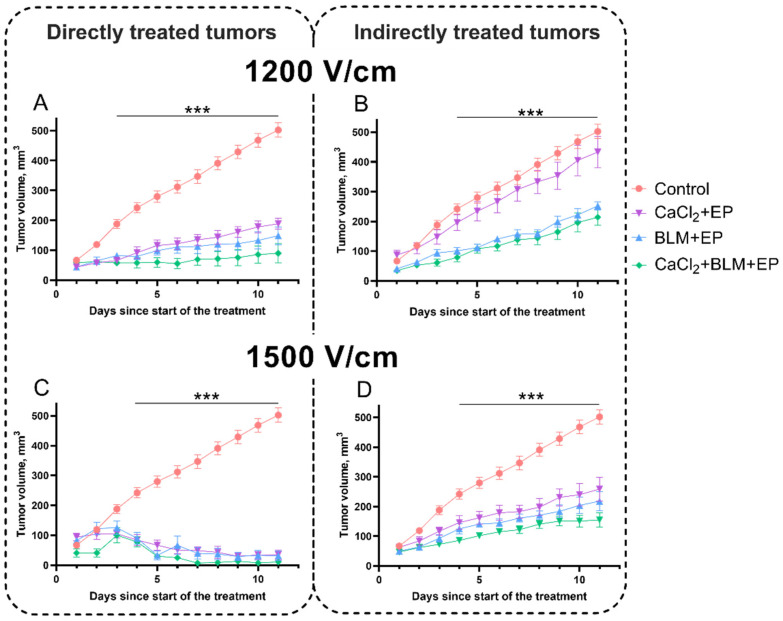
(**A**) Tumor growth changes in directly (**A**,**C**) and indirectly (**B**,**D**) treated tumors using bleomycin electrotransfer, calcium electroporation, or combination of both. Eight 1200 V/cm (**A**,**B**) and 1500 V/cm (**C**,**D**) 100 µs duration pulses were used for tumor electroporation. Bleomycin solution (200 µL, 470 µM) was injected i.v., while into CaCl_2_ solution (half of the tumor volume, 168 mM) was injected i.t. Statistical significance of the differences between experimental groups was evaluated using *t*-test. *** symbols represent statistical significance of differences between CaCl_2_ + EP, BLM + EP, and CaCl_2_ + BLM + EP compared to the control (no treatment), respectively (*p* < 0.001).

**Figure 7 cancers-14-03770-f007:**
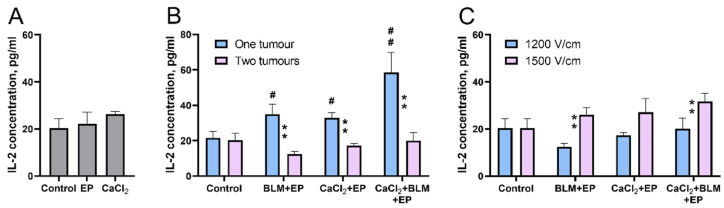
(**A**) IL-2 concentration changes in serum 10 days after tumor treatment using bleomycin electrotransfer, calcium electroporation, or combination of both. (**A**) IL-2 concentration changes after electroporation (1500 V/cm) or CaCl_2_ injection. (**B**) IL-2 concentration changes after tumor treatment using calcium electroporation, bleomycin electrotransfer, or combination of both in mice were bearing one or two tumors when 1500 V/cm electric field pulses were used for tumor electroporation. (**C**) IL-2 concentration changes after tumor treatment using calcium electroporation, bleomycin electrotransfer, or combination of both when 1200 or 1500 V/cm strength electric pulses were applied. The serum was collected at 11th day after the treatment. Statistical significance of the differences between experimental groups were evaluated using *t*-test. ** symbols represent statistical significance (*p* < 0.05 and *p* < 0.01, respectively) of differences between corresponding groups. #, ##, symbols represent statistical significance of differences between individual experimental group compared to the control (no treatment) (*p* < 0.05 and *p* < 0.01, respectively).

**Figure 8 cancers-14-03770-f008:**
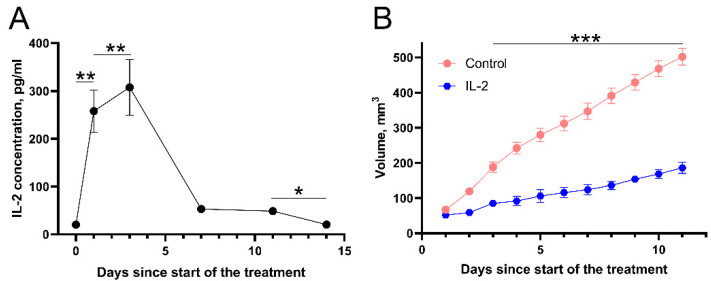
(**A**) IL-2 concentration changes in serum and (**B**) tumor growth changes after IL-coding plasmid electrotransfer into both tibialis anterior muscles during a period of 14 days. Mice bearing two tumors were used for measurement of tumor volume changes. For muscle transfection, IL-2 coding plasmid solution (1 mg/mL) was injected into both tibialis anterior muscles of each mouse and electroporated 10 min later using eight 200 V/cm, 20 ms duration pulses. Statistical significance of the differences between experimental groups was evaluated using *t*-test. The *, **, and *** symbols represent statistical significance of *p* < 0.05, 0.01, and *p* < 0.001, respectively, when compared between corresponding groups (**A**) or to the control (no treatment) (**B**).

**Figure 9 cancers-14-03770-f009:**
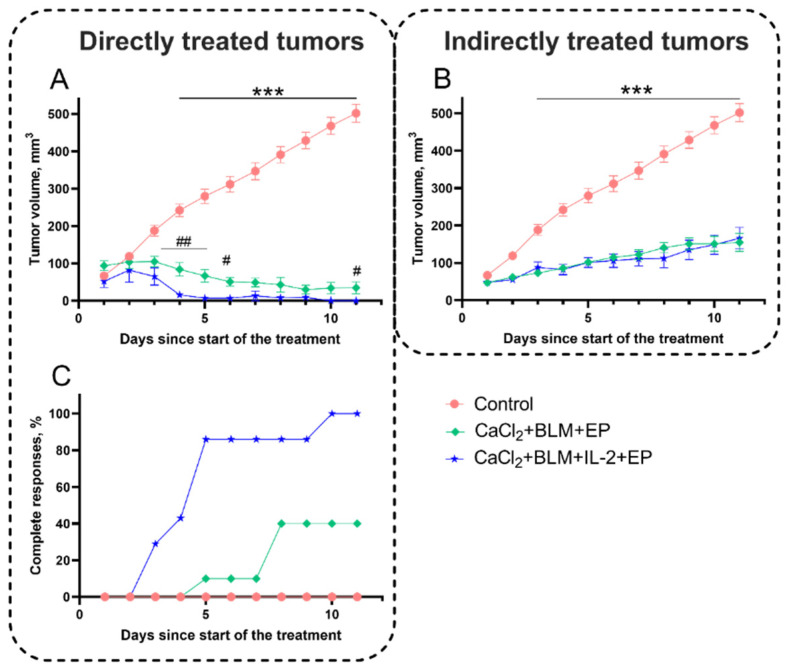
Tumor growth changes and number of complete responses after different electroporation-based therapies. Tumor growth changes in directly (**A**) and indirectly (**B**) treated tumors using combination of calcium electroporation and bleomycin electrotransfer without or with IL-2 transfection. (**C**) Percentage of directly treated tumors with complete response. Eight 1500 V/cm, 100 µs duration pulses were used for tumor electroporation. Bleomycin solution (200 µL, 470 µM) was injected i.v. while CaCl_2_ solution (half of the tumor volume, 168 mM) was injected i.t. For muscle transfection, both tibialis anterior muscles were injected with IL-2 coding plasmid solution (1 mg/mL) and electroporated 10 min later using eight 200 V/cm, 20 ms duration pulses. At least 8 mice with two tumors each were treated for every experiment point. Statistical significance of differences between experimental groups was evaluated using *t*-test. The *** symbols represent statistical significance of *p* < 0.001 between control (no treatment) and individual experimental groups. # and ## symbols represent statistical difference of *p* < 0.05 and *p* < 0.01, between Ca + BLM + EP and Ca + BLM + EP + IL-2 groups.

**Figure 10 cancers-14-03770-f010:**
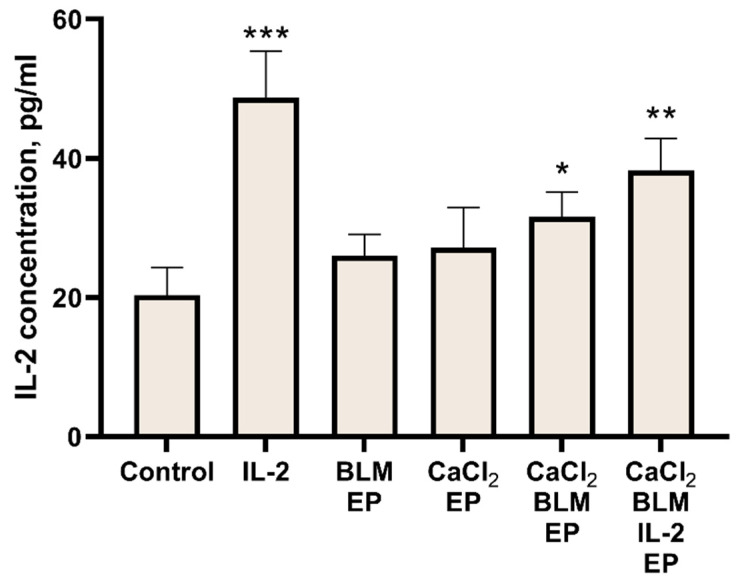
IL-2 concentration changes in serum 10 days after tumor treatment using IL-2 coding plasmid electrotransfer, bleomycin electrotransfer and calcium electroporation, or a combination of all. Mice bearing two tumors were used in all experimental groups, except IL-2 group. For muscle transfection, IL-2 coding plasmid solution (1 mg/mL) was injected into both tibialis anterior muscles and electroporated 10 min later using eight 200 V/cm, 20 ms duration pulses. Eight 1500 V/cm, 100 µs duration pulses were used for the tumor electroporation. Bleomycin solution (200 µL, 470 µM) was injected i.v. while CaCl_2_ solution (half of the tumor volume, 168 mM) was injected i.t. Serum was collected at day 11. Statistical significance of differences between experimental groups and control were evaluated using *t*-test. *, **, and *** symbols represent statistical significance of *p* < 0.05, *p* < 0.01, and *p* < 0.001, respectively, between corresponding experimental groups and control.

## Data Availability

Not applicable.

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
