# Peer review of "Induction of Bystander and Abscopal Effects after Electroporation-Based Treatments"

_cancers, 2022, doi:10.3390/cancers14153770_

Round 1

Reviewer 1 Report

This manuscript is full of strange terminologies and grammar errors that really damage the fun of reading this interesting work.

1) The manuscript must be rewritten. For instance,  there are several combinations of treatments used in the study. However,  in the manuscript, they were not mentioned clearly and exclusively such that it is difficult for readers to follow the writing of authors.

2) The informal or inconsistent terms were used too much. For instance, any difference between 'directly untreated' and 'indirectly treated'?

3) Line 72, 'resent'?

4) Line 95, 'In case of'?

5) Fig. 2B, the control group was not defined clearly. Why do we still have have BLM+EP and CaCl2+EP in the control group?

6) Any results could support that the decrease in the cell viability is due to the combination of electropration and BLM or CaCl2 instead of the increase of electric field?

7) As shown in Figs. 4 and 6, I've seen all the used combinations of treatments (CaCl2+EP, BLM+EP, and CaCl2+BLM+EP) can effectively suppress the growth of the untreated tumor. However, I cannot find any results or figures that could support the following conclusion. 

"Boosting of IL-2 serum concentration in serum did not enhance the abscopal effect of combined tumor treatment using calcium electroporation and bleomycin electrotransfer, however, it had a significant inhibitory effect on directly treated tumors. "

In summary, this manuscript may provide some interesting results of IRE and its immune responses to untreated tumors. However, the writing and presentation of the current version of manuscript cannot be accepted.

Reviewer 2 Report

In this study, Ruzgys et al. explored the induction of bystander and abscopal effects after electroporation-based treatments with bleomycin and/or CaCl2. In initial experiments, supernatants of treated 4T1 cell cultures elicited negative bystander effects on untreated cell cultures. These observations prompted the authors to investigate abscopal effects in BALB/c mice bearing two 4T1 tumors. Importantly, treatment of one tumor inhibited the growth of the second tumor, suggesting involvement of immune system responses. This hypothesis was tested by monitoring IL-2 in mice sera and by increasing IL-2 via genetic overexpression in the mice. IL-2 was elevated in sera of mice when electroplated with bleomycin/CaCl2 combination. Overexpression of IL-2 following combined CaCl2/bleomycin electroporation resulted in 100% complete response of the directly treated tumors. However, there was no additional effect with IL-2 overexpression on the indirectly treated tumors.

Overall, these findings are highly relevant to the field and suggest that treatments based on electroporation of bleomycin and/or CaCl2 can have abscopal anti-tumor effects in vivo. Direct treatment of a tumor may stimulate the immune system to elicit additional anti-tumor responses. Electroporation-based treatments may be combined with IL-2 therapy to enhance complete responses.

I have some questions/discussion points, and suggestions improving the method section, statistical analysis, and figure legends.

1) The abscopal effect with electroporation seems not to be associated with immune responses mediated by IL-2. In order to verify that IL-2 is associated with direct (or maybe also with indirect) anti-tumor effects is it feasible to block IL-2 in the BALB/c mice during electroporation treatments?

2) In the initial experiments, bystander effects were found in untreated cells. Since the immune system is absent in the cell cultures, what is the mechanism of the bystander effect? Is the bystander effect restricted to the same tumor cell type only or did you see effects also on other tumor cells or even on healthy non-tumor cells?

3) Figure legends of Figure 2 and 3 need to be improved: Indicate whether the data points represent means of a number of cell cultures each condition of a representative experiment or represent the means of a number of independent experiments

4) Statistical analysis need to be included in Figure 2 and Figure 3. All other Figures need some clarification about between which groups and time points statistical significance was indicated. To my knowledge, t-test can only be done between two conditions in one graph. In the Figures, more than two conditions, time points or groups are plotted in one graph, and in this case ANOVA and multiple comparison test may be the better choice

5) Figure 7: is too small, text is hardly readable

6) Figure 7B and C: induction of IL-2 concentration in sera of treated mice with one tumor is not statistical significant different to untreated controls? check statistical analysis

7) Legend Figure 8 B: include ***symbols represent statistical analysis of p<0.001 between control and IL-2 at time point day xy etc

8) Figure 8B and Figure 10: was this done in mice bearing one or two tumors, clarify in the figure legends

9) line 91: add information about the 4T1 cell line, cell type and origin etc.

10) line 133: a detailed flow cytometry protocol and used equipment should be included in the method section

11) line 153: add information about the BALB/c mice strain type, whether the mice have been purchased or own breeding, housing conditions (SPF etc), sexes, and age of mice at start point of experiment

12) line 156, 174: add what physiological solution has been used

13) line 167: add information about the volume (µL) of CaCl2 solution injected directly in the tumors

14) line 174: include a detailed description of the used IL-2 coding plasmid

15) line 190: add the method by which the length, width and height of the tumors was measured

16) line 200: add the used ELISA reader equipment

Reviewer 3 Report

The article titled “Induction of bystander and abscopal effects after electroporation-based treatments” by Ruzgyset al. presents a set of fascinating study involving calcium electroporation, bleomycin electrotransfer and IL-2 gene delivery on an 4T1 cancer model. The observation of bystander effect and abscopal effect are of particular interest, as the authors clearly demonstrate that, using both in vitro and in vivo models that such effects are inducted with electroporation (EP). The reviewer also looks forward to seeing the authors’ inspiring presentation at the 4th World Congress on Electroporation in October 2022.

There are 3 closely related studies being presented in this manuscript: (1) in vitro bystander effect, (2) in vivo abscopal effect and (3) enhanced treatment by IL-2 coding plasmid electrotransfer. While the reviewer finds all of them convincing with the results presented, the reviewer would raise the following major concerns regarding the completeness of all 3 studies.

(1) Lack of explanation on the in vitro bystander effect

Despite the authors argue that “Synthesis and release of damage-associated molecular patterns (DAMPs) [30,36,37] following electroporation is the most common explanation of how electroporation-based cancer treatments can induce immune responses.” In the introduction, the reviewer does not find any analysis of the DAMS in the conditioned medium (CM), to explain the results presented in Figure 3.

(2) Lack of explanation on in vivo abscopal effect in the “two tumors in one mouse” model

Even though the authors argue the importance of IL-2 in the discussion, the measured changes of IL-2 have little correlation with the magnitude of abscopal effect observed by the authors. There is little in-depth examination of the immune system of which researchers usually believe to contribute to the abscopal effect.

(3) No analysis on why IL-2 “had a significant inhibitory effect on directly treated tumors”. IL-2 is known to have therapeutic effects in the 4T1 model, does it offer additional suppression on tumor growth, or can it be added synergistically to the EP therapy?

(4) In addition to the 4T1 model, does the bystander and/or abscopal effect still hold true for other cancer models? 

Other minor concerns and suggestions:

·            The words “electroporation-based treatments” in the title can be overly generalized. The reviewer would recommend specifying the actual electroporation-based treatments the authors have studied.

·            The composition (or ingredients) of the conditioned medium (CM) would need to be further clarified. Does it have any Calcium of BLM?

·            Please explain why “medium taken from untreated cells (control) affected cell viability to some extent”. The medium is just regular cell culture medium without any additives that is not supposed to cause cell death, therefore affect viability. 

·            Fig 3, recommend rearrange the plots. The bar plots of “CM from BLM + EP 50%” and “CM from CaCl2 + EP 50%” each appear twice.

·            Fig 4 and 5.  Why is there a hump of tumor volume curve on day 4 in the BLM group?

·            Fig 5, the colors of the curves are too similar, especially when they are overlapping.

·            How EP conditions are decided, especially 1200 vs 1500 v/cm in Fig 4 and Fig5?

·            Fig 6A, a large number of the “directed treated tumors” have reached over 100 mm3 by day 11, while in Fig 4, almost all tumors are below 100 mm3 at day 11. The EP condition, Bleomycin and CaCl2 dosages seem to remain the same. Please explain.

·            Fig 6, the “control” group would need to be clarified. Is it “no treatment”?

·            Fig 7, caption is confusing, especially for A and B. What are the tumor conditions (one vs two) and what are the EP conditions?

·            Fig 10, looks like Duplicated data with fig 7B for “BLM + EP”, “CaCL2 + EP” and “BLM CaCl2 + EP” group. Can these IL-2 data be rearranged?

·            Page 11. “Since boost of serum IL-2 concentration following IL-2 plasmid electrotransfer resulted in significant abscopal effect on indirectly treated tumors”. The reviewer does not find results supporting this claim.

Round 2

Reviewer 1 Report

No further comments.

Reviewer 3 Report

The reviewer appreciates the effort in addressing the comments and revising the manuscript. In particular, the reviewer welcomes the added clarity and the discussion on IL-1 in the last paragraph in the discussion.

While most of the responses are satisfactory, the reviewer still encourages the authors to provide more insightful perspectives on the mechanism of bystander effect, as such effect has been reported by the same group (Ruzgys et. al. 2021).

The reviewers would also encourage the authors to be specific on the used electroporation methods as they appear in the title, rather than using generic terms as “electroporation-based treatments”

The reviewer would discourage presenting duplicated data on separate figures. To help guide readers, a larger figure with more subplots can be used.  

Finally, even though IL-2 EP has shown to improve the abscopal effect, as shown in Fig 9. The reviewer does not agree that IL-2 plasmid electrotransfer or IL-2 transfection has an abscopal effect, as described in Fig 8 and 1st paragraph on page 12. The IL-2 EP is not directed on a tumor, but more of a systemic therapy. In fact, neither the IL-2 coding plasmid, nor the EP were applied on a tumor.

As described, “Electrotransfection was done on both tibialis anterior muscles of each treated mouse.”  and “The induction of the tumor was done by subcutaneous injection of 4T1 cells on one or two dorsal caudal sides of the mice.”.